# Molecular Pathways of Genistein Activity in Breast Cancer Cells

**DOI:** 10.3390/ijms25105556

**Published:** 2024-05-20

**Authors:** Evangelia K. Konstantinou, Aristea Gioxari, Maria Dimitriou, George I. Panoutsopoulos, Athanasios A. Panagiotopoulos

**Affiliations:** Department of Nutritional Science and Dietetics, School of Health Sciences, University of the Peloponnese, Antikalamos, 24100 Kalamata, Greece; e.konstantinou@go.uop.gr (E.K.K.); a.gioxari@go.uop.gr (A.G.); m.dimitriou@go.uop.gr (M.D.); gpanouts@go.uop.gr (G.I.P.)

**Keywords:** genistein, breast cancer, natural products, flavonoids, antioxidants

## Abstract

The most common malignancy in women is breast cancer. During the development of cancer, oncogenic transcription factors facilitate the overproduction of inflammatory cytokines and cell adhesion molecules. Antiapoptotic proteins are markedly upregulated in cancer cells, which promotes tumor development, metastasis, and cell survival. Promising findings have been found in studies on the cell cycle-mediated apoptosis pathway for medication development and treatment. Dietary phytoconstituents have been studied in great detail for their potential to prevent cancer by triggering the body’s defense mechanisms. The underlying mechanisms of action may be clarified by considering the role of polyphenols in important cancer signaling pathways. Phenolic acids, flavonoids, tannins, coumarins, lignans, lignins, naphthoquinones, anthraquinones, xanthones, and stilbenes are examples of natural chemicals that are being studied for potential anticancer drugs. These substances are also vital for signaling pathways. This review focuses on innovations in the study of polyphenol genistein’s effects on breast cancer cells and presents integrated chemical biology methods to harness mechanisms of action for important therapeutic advances.

## 1. Introduction

One of the most prevalent and fatal diseases in the world today is cancer, with breast cancer (BC) accounting for the majority of diagnoses among women [1,2]. To stop or delay the stages of carcinogenesis, there are three basic approaches [2,3,4,5,6,7,8,9,10]. The main strategy under consideration is a preventive one that suppresses both the mutagenic and toxic effects, thus preventing the beginning and promotion of tumors [4,5]. Through a variety of mechanisms, including hormone regulation, signal transduction control, angiogenesis inhibition, antioxidant mechanisms, and immunity modulation, the secondary strategy offers anticancer potential during the early stages of cancer genesis [4,5]. Ultimately, this prevents the cancer from progressing [4,5]. The third approach to treating and preventing cancer is suppressing the tumor’s ability to spread by upregulating genes that prevent metastasis, protecting the extracellular matrix from deterioration, and controlling cell adhesion molecules [4,5].

The complex nature of the multistep tumorigenic process arises from its initiation, development, and progression [11,12]. The entry and spread of carcinogenic substances into the cell, particularly within the nucleus, as well as their interaction with DNA, which ultimately leads to mutagenesis and the appearance of toxic effects, are all recognized as forms of initiation [4,13,14,15,16,17]. Numerous oncogenic transcription factors contribute to the overproduction of inflammatory cytokines and cellular adhesion molecules during the development of cancer [5,15]. Additionally, antiapoptotic proteins are significantly upregulated in cancer cells, whereas proapoptotic protein pathways are significantly downregulated [13,14]. Improvements in cell survival, metastasis, and tumor formation are caused by these modifications in protein expression [5,15]. The increased production of the proteins known as inhibitors of apoptosis, which are important in programmed cell death, also contributes to chemoresistance [13,18,19,20,21]. Changes in apoptosis are the cause of tumor growth and metastasis, as well as tumor resistance to treatment [13,14,20,22,23,24,25,26].

At present, natural products and food-based polyphenols—also known as nutraceuticals, functional foods, dietary supplements, and phytochemicals—are being developed for their potential health advantages [27,28,29,30,31,32,33,34,35,36,37,38,39,40,41,42,43]. The quantity and bioavailability of polyphenols determine their impact on health [9,22,37,38,44,45,46,47,48].

With regard to their great structural and chemical diversity, a wide range of natural compounds and their derivatives are being studied as potential cancer prevention and therapy agents [27,29,30,31,32,33,34,35,36,37,38,39,49,50,51]. Natural products are essential for signaling pathways because they modulate a broad variety of enzymes and cell receptors [40,52]. The use of complementary and alternative medicine, as well as dietary strategies to prevent and treat BC, is becoming more and more popular [48,53]. The need for more effective therapeutic and preventive measures with fewer side effects is urgent, and the likelihood of accomplishing this aim is increased when screening natural compounds derived from plants [41]. Numerous natural chemicals produced from plants, such as the widely recognized phenolic acids, flavonoids, tannins, coumarins, lignans, lignins, naphthoquinones, anthraquinones, xanthones, and stilbenes, are being researched for future medication development and enhancement [11,12,27,28,29,30,31,32,33,34,36,49,50,51,54,55,56,57,58].

In this review, we highlight current developments in the investigation of natural products as effective anticancer medicines, with a particular emphasis on the impact of the well-known polyphenol genistein on BC cells. We provide an array of integrated chemical biology approaches designed to leverage pertinent mechanisms of action that could result in significant therapeutic advances for the management of BC.

## 2. History of Genistein

Genistein (5,7-Dihydroxy-3-(4-hydroxyphenyl)-4H-1-benzopyran-4-one or 4′,5,7-Trihydroxyisoflavone) (C_15_H_10_O_5_) is a naturally occurring compound that structurally fits into the isoflavonoids class of substances [59,60,61]. It is classified as a phytoestrogen and an inhibitor of angiogenesis [59,60,61]. The chemical name comes from its initial isolation in 1899 from *Genista tinctoria*, also known as Dyer’s broom [59,61]. The compound structure was developed in 1926 when it was discovered that it shared the same structure as prutenol. In 1928, it was chemically produced [59,60]. Genistein is a principal secondary metabolite of Glycine max’s and Trifolium species’ [60].

The 3-phenylchromen-4-one nucleus of this well-known plant secondary metabolite is composed of two aromatic rings, A and B. Moreover, Figure 1 shows that these rings are connected to another carbon pyran ring (C). In addition, there is an oxo group at ring C’s C4 location and a double bond between C2 and C3 in its basic carbon skeleton. In addition, rings A and B have three hydroxyl groups at positions C5, C7, and C4, respectively.

Isoflavonoids belong to a large and varied class of substances called phenolic compounds that are produced by plant metabolism [62,63]. They are well-known for several advantages. Natural polyphenolic compounds, phenolic compounds from plant origin, are produced by plants as a protective mechanism against environmental stressors [62,63]. The most well-known use of genistein and related phytoestrogens is as a chemopreventive medication for a range of malignancies and illnesses in humans [64]. Phytoestrogens are plant-based compounds that share a similarity in structure with the primary mammalian estrogen, 17β-estradiol (E2) [64,65]. These compounds are known to potentially selectively modulate estrogen receptors. Genistein, one such phytoestrogen, has a significantly greater affinity for binding to ERβ than to ERα [64,65]. Specifically, its affinity for ERβ is 30 times higher than for ERα [64,65]. This difference in binding affinity may have implications for the risk of breast cancer in cells with varying ERα/ERβ ratios [64]. For many generations, Asian populations have consumed large amounts of soybeans and meals derived from them, seemingly without any negative consequences [64]. However, because of the isoflavonoids’ other estrogenic properties, there is concern about possible negative effects. Isoflavonoids, which include glycitein, daidzein, and genistein in their conjugated forms, share structural similarities with estradiol and possess estrogenic properties [64]. There is evidence that isoflavonoids may exert infertility in grazing animals, caged cheetahs, and California quail, animals in which the reproductive effects of isoflavonoids were noticeable [64]. However, the affinity of isoflavonoids for the estrogen receptor is 100–1000 times lower than that of estradiol [64].

Several studies supporting the multi-target mechanism of genistein’s anti-tumor activities have been conducted [66,67,68,69,70,71,72,73,74]. Cell cycle regulation [75,76], tyrosine kinases [77], DNA topoisomerases [78], telomerase [79], apoptosis [80], and angiogenesis [81] have all been found to be inhibited by genistein.

Genistein can be organically synthesized using deoxybenzoin as a substrate [60]. So far, a multitude of methods for the synthesis of genistein and its analogs, using the Suzuki–Miyaura coupling reaction of 3-iodochromone with the suitable boronic acid or aryl boronic ester, have been published [82,83,84,85,86,87,88].

## 3. Natural Occurrences of Genistein

The majority of flavonoids are glycosidic conjugates found in plant cell vacuoles [63,89,90,91]. After heating, the labile malonyl glucosides of genistein and daizenin in soybeans are quickly converted to non-acylated glucosides [63,89,90,91]. Legumes are the primary source of genistein, but it can also be found in other groups (Table 1). Soybean total isoflavone concentrations vary widely depending on the cultivar and growth conditions, ranging from 1161 to 2743 μg/g [92,93]. It was demonstrated that there was a negative correlation between genistein content and plant height, days of maturity, and yield [92,93]. When combined as free and conjugated forms, genistein constitutes 50–70% of the total isoflavone content in soybeans, making it the most prevalent isoflavone [92,93]. Though its free form shows high physiological activity, genistein conjugates, especially 7-O-β-glycoside, are the most abundantly represented forms of this isoflavonoid in plant tissues [92,93]. There are numerous genistein sources recommended as soybean substitutes [92,93]. One of the major medical cash crops cultivated in many nations is *Withania somnifer* L. (*Solanaceae*) [94,95,96]. Significant amounts of genistein were produced by *Pueraria candollei* leaf, stem, and root explants [97]. Genistein may be obtained from *Psoralea corylifolia* L. callus cultures [98]. If the culture parameters (temperature, light, and medium composition) are properly adjusted, cell suspension cultures and callus cultures may have higher quantities of genistein [98]. The process of isolating genistein from plant source materials is becoming more and more sophisticated [99,100].

## 4. Biosynthesis of Genistein

According to studies, phenolic chemicals may help lower the risk of developing long-term conditions such as diabetes, cancer, heart disease, and neurological disorders [56,134]. Many different types of life forms (bacteria, fungi, and plants) produce aromatic chemicals through the shikimic acid pathway of biosynthesis [56,134]. The nutritional requirements of animals and humans for shikimate-inferred aromatic amino acids (i.e., essential amino acids) demonstrate their deficiencies in this approach [56,134]. Many naturally occurring phenolic compounds, including lignans, flavonoids, coumarins, and cinnamic acids, include the crucial C6-C3 structure of phenylpropane, which is formed from the amino acids phenylalanine and tyrosine [18,135,136]. Numerous alkaloids production begins with these amino acids and tryptophan [18,135,136].

Leguminosae plant species are primarily characterized by their isoflavonoids, which include genistein [137]. The phenylpropanoid pathway’s intermediary substrates, liquiritigenin and naringenin, are utilized to create isoflavonoids [137]. The majority of plants contain naringenin, which may also be the source of other phenylpropanoid pathway chemicals such as flavonoids, flavonol, and anthocyanin [137].

Following the processes involving the transfer of shikimic acid via chorismic acid to L-phenylalanine (L-Phe), L-Phe is converted to 4-hydroxycinnamoyl-CoA (p-coumaroyl-CoA) (Figure 2) [138,139]. The phenylpropanoid (PP) route is initiated by the condensation of this molecule with three molecules of malonyl-CoA in the presence of the enzymes chalcone synthase, naringenin-chalcone synthase, and chalcone isomerase, leading to the production of naringenin [138,139].

After that, 2-hydroxyflavanone synthase transforms naringenin by abstracting the H-radical at C3 and moving ring B from C2 to C3 in the presence of oxygen and NADPH. Since it is strictly stereoselective, (2R)-flavanones cannot be acted upon by this enzyme [138,139]. The unstable 2-hydroxyisoflavanone receives the generated C2 radical [138,139]. A particular dehydratase enzyme subsequently catalyzes the process of dehydration, which results in the formation of the isoflavonoid genistein [138,139].

From its initial synthesis in 1928 until today, many methods involving the laboratory synthesis of genistein and its analogues have been developed [59,60,82,83,86,97,140,141]. Trihydroxybenzoin, which is obtained by acylating phloroglucinol with substituted phenyl acetonitrile using HCl and anhydrous ZnCl_2_ in dry ether as a catalyst, is treated to provide genistein [86]. By preserving two hydroxyl substituents in triol as methoxymethyl ester, genistein has been synthesized from 2,4,6-trihydroxyphenyl ethanone in order to avoid the problem of dimethoxymethyl dimethylamine interacting with phenol [83]. Ferrier rearrangements of 3,4-di-O-acetyl-L-rhamnal with 3-bromopropanol can also yield derivatives of genistein, such as 2,3-unsaturated bromoalkylglycosides, which are then epoxidized with meta-chloroperoxybenzoic acid and then linked to genistein [140]. Additionally, novel chemical glycosylation and glycoconjugation procedures are employed to generate derivatives of genistein [141].

## 5. Effects of Genistein in BC

### 5.1. Induction of Apoptosis

The chemotherapeutic drug genistein has alone been thoroughly investigated for use in the treatment of cancer. It controls the angiogenesis, apoptosis, cell cycle, and metastasis. An increase in cell division and a decrease in programmed cell death are associated with tumor growth. Numerous factors can initiate apoptosis, as per recent in vitro studies. Genistein causes apoptosis in several BC cell lines.

One potential molecular mechanism for the prevention of mammary cancer has been suggested to be the stimulation of the peroxisome proliferator-activated receptor gamma (PPARγ) pathway (Figure 3) [142]. This pathway includes cyclin B1, PTEN, and PPAR [142]. When genistein was added to MDA-MB-231 cells along with arachidonic acid, docosahexaenoic acid, and eicosapentaenoic acid, PPAR expression was upregulated while cyclooxygenase-2 and prostaglandin E2 expression was reduced, reversing invasiveness in the cancer cells [142]. By decreasing inflammatory prostanoids and COX-2 activity, as well as changing cell signaling, N-3 polyunsaturated fatty acids (PUFA) and genistein have been associated with a lower risk of cancer [142]. According to a study, genistein and n-3 PUFA can suppress COX-2 expression, which lowers the production of prostaglandin E2 (PGE2) in MDA-MB-231 human BC cells [142]. The elevated levels of PGE2 and cell invasiveness were reversed by genistein in conjunction with arachidonic acid (AA), eicosapentaenoic acid (EPA), and docosahexaenoic acid (DHA) [142].

In MDA-MB-231 and BT-474 cells, apoptosis was seen as a result of genistein’s synergistic action when treated with anti-BC medications, lowering their chemoresistance [143,144].

A study by Satoh et al. investigates the possibility of using genistein, a vital component of soybean isoflavone, as an anticancer agent against HER-2-overexpressing BC cells [143]. It was discovered that genistein, at low doses, increased the cytotoxic effect of adriamycin (ADR), mostly as a result of an increase in necrotic-like cell death [143]. BC cells’ human epidermal growth factor receptor 2 (HER2) and protein kinase B (Akt) were remarkably inactivated when genistein and ADR were combined, suggesting that genistein promotes necrotic-like cell death in BC cells [143].

Tamoxifen is a treatment and preventive option for estrogen receptor-positive BCs, but around 40% of these tumors are resistant to tamoxifen [144]. The overexpression of the HER2 gene is associated with tamoxifen resistance, and tamoxifen activity increases in the presence of a reduced HER2 expression [144]. Genistein, an isoflavone found in soy, has anticancer effects and inhibits the expression of HER2 and Erα [144]. A study by Mai et al. investigates the hypothesis that genistein could increase the susceptibility of ER+ and HER2-overexpressing BC cells to tamoxifen treatment [144]. Tamoxifen and genistein, given together, have a synergistic effect on the growth of BT-474 human BC cells in vitro. Nevertheless, further in vivo research is needed to validate the impact of the genistein and tamoxifen combination on human BC, even if the above study may provide a novel approach to treating and preventing tamoxifen-insensitive/resistant BC [144].

Furthermore, apoptosis may be triggered by the cell-deadening enzymes calpain and caspase, which are activated by calcium ions [145]. The reduction of calcium storage in the endoplasmic reticulum, elevated Ca^2+^ levels, calpain activation, and the obstruction of calpain’s Ca^2+^ binding sites lead to an enhanced cytosolic Ca^2+^ buffering capability. Additionally, caspase suppression diminishes apoptosis in cancerous cells [145]. Therefore, one method by which genistein induces apoptosis is through its ability to regulate the level of Ca^2+^ in cells [145]. A study by Sergeev et al. found that genistein induces apoptosis in BC cells by activating Ca^2+^-dependent proapoptotic proteases, l-calpain, and caspase-12 (Figure 3) [145]. Genistein treatment of MCF-7 BC cells resulted in a prolonged rise in intracellular Ca^2+^, which was linked to the activation of caspase-12 and l-calpain [145]. According to the study, genistein in BC cells may target Ca^2+^-dependent proteases, and its apoptotic mechanism is driven by cellular Ca^2+^ regulatory activity [145]. The findings are consistent with the theory that steroid hormones and their analogs frequently and effectively induce apoptosis through a prolonged rise in [Ca^2+^]_i_ [145]. For example, the selective stimulation of Ca^2+^-mediated cell death in cancer or the targeted shielding of neurons from Ca^2+^ cytotoxicity in degenerative illnesses can both be accomplished by utilizing the Ca^2+^-mediated apoptotic pathway [145]. Immunocompromised animals, as well as in vitro and in vivo models of MDA-MB-435 and Hs578t cells, demonstrated mammary tumor development through cell viability impairment and ultimately cell death [145].

A study by Zhao et al. explores the regulation of the cancerous inhibitor of protein phosphatase 2A (CIP2A), a novel oncogene often overexpressed in BC [146]. It was found that genistein induces the downregulation of CIP2A in MCF-7-C3 and T47D BC cells, which is linked to its growth inhibition and apoptotic activities [146]. The downregulation involved both transcriptional suppression and proteasomal degradation [146]. The study also discovered that CIP2A’s downregulation is influenced by modifications to E2 promoter binding factor 1 (E2F1)-mediated transcriptional regulation [146]. This set of data supports the idea of genistein-associated anti-tumor activities and their implications for BC prevention and treatment [146]. The research lends credence to the notion that CIP2A could be a useful target for the creation of innovative anticancer drugs [146].

A study by Shao et al. investigated if genistein had other suppressive effects on BC progression [147]. They found that genistein inhibited the in vitro invasion of MCF-7 and MDA-MB-231 cells, downregulating matrix metalloproteinase-9 (MMP-9) and upregulating the tissue inhibitor of matrix metalloproteinase-1 (MMP-1) [147]. In vivo studies showed that genistein inhibited tumor growth, stimulated apoptosis, and upregulated p21^WAF1/CIP1^ expression [147]. p21^WAF1/CIP1^ was first discovered as a 21 kDa protein that prevented the activation of the cyclin/CDK (Cip1) complex [148]. Later, it was discovered to be an overexpressed gene at 6p21.1 (sdi1) in senescent cells [148]. Additionally, the p21 gene product is transcriptionally activated by p53 when DNA damage occurs (WAF1) [148]. Genistein also inhibited angiogenesis by decreasing vessel density and th vascular endothelial growth factor (VEGF), and transforming growth Factor beta 1 (TGF-β1) [147].

Consequently, genistein has been well documented as causing cancer cells to undergo apoptosis through several methods, such as cell-signaling pathways. Evidence of genistein’s apoptotic properties on BC cells, both in vitro and in vivo, is emphasized, indicating genistein’s potential utility. To be utilized as a therapeutic medication, genistein may need more study to identify its intracellular targets.

### 5.2. Mechanism of Cell Cycle Arrest and Anti-Proliferative Effects

The natural protein tyrosine kinase inhibitor genistein causes G2/M arrest and apoptosis, which has an anticancer impact [149]. To represent genistein-inhibited phosphotyrosine cascades, a study by Yan et al. merged tyrosine phosphoprotein enrichment with MS-based quantitative proteomics technology to identify genistein-regulated tyrosine phosphoproteins worldwide [149]. This study identified genistein-regulated tyrosine phosphoproteins on 181 genistein-regulated proteins, revealing new inhibitory effectors with no previously known function in the anticancer mechanism of genistein [149]. The phosphoproteins inhibit the tyrosine kinases EGFR, PDGFR, insulin receptor, Abl, Fgr, Itk, Fyn, and Src [149]. Core signaling molecules inhibited by genistein can be categorized into the canonical Receptor-MAPK or Receptor-PI3K/AKT cascades [149].

Flavonoids target various signaling pathways, including apoptosis, cell cycle arrest, mitogen-activated protein kinase (MAPK), phosphoinositide 3-kinase (PI3K)/AKT kinase, and metastasis. Polo-like kinase 1 (PLK1) is a valuable target in cancer treatment due to its prognostic implications and clinical relevance. Recent in vitro and in vivo studies suggest that flavonoids, including genistein, directly inhibit PLK1 inhibitory activity, with future research focusing on its anticancer effects [150,151]. Extensive investigations have shown that genistein has significant inhibitory activity at nearly every step of the metastatic cascade [150,151]. At high concentrations, it can inhibit the proteins involved in primary tumor growth and apoptosis, including the cyclin class of cell cycle regulators and the Akt family of proteins [150,151]. It can prevent cancer cells from dividing, migrating, and invading at lower concentrations by blocking the transforming growth factor (TGF)-beta signaling pathway [150,151]. Genistein has been shown to inhibit human cancer metastasis and modulate metastatic potential markers [150,151].

Genistein has been shown to exhibit anti-proliferative effects, including the blocking of NF-kB pathways and the consequent activation of NF-kB [150,151]. Modulation of the EGFR/Akt/NFκB pathway contributes to cell differentiation [152], which ultimately results in the cancer cells apoptosis. Genistein inhibits Akt activity, which facilitates the deactivation of downstream signaling pathways such as NF-κB [153]. The electrophoretic mobility shift test in MDA-MB-231 cells, and the suppression of Akt activation by blocking the triggering of the EGF signal, were used to demonstrate this [153].

A study by Pavese et al. investigated the effects of genistein, present in soybeans, on suppressing inflammatory responses in the mammary glands of mature female rats treated with topical tetracyclopropane (TPA) [154]. The combined effects of genistein and capsaicin on COX-2, pJNK, pERK, and pp38 expressions were found to be additive or nonadditive [154]. MCF-7 BC cells demonstrated the synergistic impact of capsaicin and genistein in vitro. The study also discovered that capsaicin and genistein together have anti-inflammatory and anticarcinogenic effects by modifying COX-2 and AMPK, as well as potentially a number of other mitogen-activated protein kinases, in a nonsynergistic or synergistic manner [154]. Genistein pretreatment inactivates NF-κB and may contribute to increased growth inhibition and apoptosis induced by cisplatin, docetaxel, and doxorubicin in prostate, breast, lung, and pancreatic cancer cells [155]. Genistein suppresses the protein levels of MEK5, total ERK5, and phospho-ERK5, which are consistent with the inhibition of cell growth and the induction of apoptosis [155]. The inhibition of the MEK5/ERK5/NF-κB pathway may be an important mechanism by which genistein suppresses cell growth and induces apoptosis [155].

The Akt pathway partially regulates the inactivation of NF-κB cancer cells, a process that is aided by the MEK5/ERK5 pathway [155,156]. These results have been validated in silico investigations, where lysine, serine, and aspartic acid amino acid residues have been identified as key contributors [157]. Proliferation may be inhibited by deactivating the Akt pathway [158]. A rise in sub G(0)/G(1) apoptotic fractions was seen in MCF-7 and MCF-7 HER2 cells, which might be attributed to the extrinsic programmed cell death pathway being triggered, p53 being upregulated, IB being less phosphorylated, and p65 evading nuclear translocation [159,160]. At the G2/M phase, genistein stops the cell-division cycle, resulting in arrest [161]. Genistein, mediated by mitogen-activated protein kinase, which subsequently represses cyclin B1 and Cdc25C and elevates c-Jun and c-Fos levels, is linked to cell division arrest at the G2/M phase [162,163].

### 5.3. Reducing Angiogenesis

BC progression is linked to the degradation of the extracellular matrix by metalloproteinases (MMPs), which affect the growth and invasiveness of cancer cells [164]. Genistein inhibits the growth of various cancer cells in vitro [164]. A study by Kousidou et al. examined the expression of mRNAs encoded for MMPs and their endogenous inhibitors (TIMPs) associated with BC cell pathogenesis and metastatic potential [164]. Gene expression was examined in cell cultures of BC cell lines, including MDA-MB-231 and MCF-7 [164]. The addition of genistein resulted in the downregulation of MMP gene transcription in MDA-MB-231 and most MMPs in MCF-7 cells [164]. This inhibitory effect on MMPs was functionally confirmed, as it significantly reduced cancer cell invasion properties [164]. The results suggest genistein may be valuable in preventing BC cell metastasis, as it acts as both a transcriptional modulator of genes involved in this pathogenetic process and a suppressor of invasiveness [164]. Another study by Latocha et al. verified that genistein addition in T47D cells results in a decrease in the expression of MMPs 2, 3, 3, and 15, preventing angiogenesis and metastasis [165].

Mukund et al. conducted a study to examine the possibility of therapeutic suppression of hypoxia-inducible factor-1α (HIF-1α) action in managing diseases, including BC [166]. Genistein was found to downregulate HIF-1α in BC cell lines [166]. The research also revealed that genistein binds to the FIH-1 binding site of the HIF-1α protein, suggesting that genistein and/or HIF-1α antagonists could be a potential treatment for BC [166].

Moreover, in silico research has demonstrated the role of Akt, HIF1α, and VEGF cascades in genistein-induced angiogenesis inhibition [167]. In addition, researchers created lipo-polymer hybrid nanoconstructs bound to spermine. These systems worked together to disperse genistein and BC drugs. This prevented the calcifications of the mammary artery [167]. These findings could pave the way for the creation of brand-new chemotherapeutic medication combinations that incorporate nanoparticle technology and anti-angiogenic genistein [167].

### 5.4. Actions on Cancer Stem Cells

BC stem cells (BCSCs) have the ability to self-renew and differentiate. The estrogen receptor-negative (−) BCSCs are affected by differentiated ER-positive (+) tumor cells through paracrine signaling [168]. Genistein can act on ER+ BC cells [168]. A transwell co-culture system was used to analyze the interaction between ER+ and ER− BCSCs [168]. In vitro results showed that the genistein concentration at 2 µM and 40 nM promoted the morphological alteration of mammospheres, reduced the ratio of CD44^+^/CD24^−^/ESA^+^ cells, and upregulated the expression of differentiated cell markers [168]. This suggests genistein can induce BCSC differentiation through a paracrine mechanism [168]. Amphiregulin produced from ER+ cancer cells activates the MEK/ERK and PI3K/Akt signaling pathways, which are linked to the differentiation-inducing impact of genistein on mammospheres [168].

According to a study by Montales et al., the Akt inhibitor perifosine increased the expression of tensin homolog deleted on chromosome ten (PTEN) and tumor suppressor phosphatase, simulating genistein’s prevention of mammosphere formation [169].

Female mice’s breast tumor suppressors PTEN and E-cadherin expression were elevated and their mammary adiposity was decreased after post-weaning dietary exposure to soy protein isolate and its bioactive isoflavone genistein [170]. The effects of genistein during the development of mouse mammary stromal fibroblast-like (MSF) cells into adipocytes was assessed using SV40-immortalized cells [170]. These results indicate a molecular mechanism supporting genistein’s direct control of mammary adiposity for the prevention of BC [170]. It was discovered that ERβ signaling mediates adipocyte differentiation through a linear pathway that includes PPARγ and ERβ activation [170].

Genistein increased apoptosis and slowed the growth and multiplication of MCF-7 BC cells [171]. Through the downregulation of the Hedgehog-glioma-associated oncogene homolog 1 (Hedgehog-Gli1) signaling pathway, genistein suppressed BC stem-like cells and lowered BC stem cells both in vitro and in vivo [171].

### 5.5. Gene Regulation and miRNA Εxpression

It has been discovered that genistein affects BC via regulating genes, specifically, up- and downregulating the genes involved in cell salvage [172]. Genes related to the stress response, transcription, and enzymes in the salvage pathway were all increased, suggesting a role for genistein in the activation of the salvage response [173]. Molecular chaperones, another name for heat shock proteins, are thought to play a key role in a cell’s ability to adapt to changes in its environment [172]. The inhibition of ER- and insulin-like growth factor-arbitrated pathways in MCF-7 cells, caused by the dysregulation of SRF expression, is the cause of genistein’s inhibitory action [174,175].

Subsequently, it is believed that genistein affects gene transcription and controls epigenetic processes [172]. When genistein was given to adult female rats throughout the conception process, the BRCA1 gene’s CpG methylation decreased [172]. This was demonstrated by a decrease in the expression of CYP1B1, a potential target for the aryl hydrocarbon receptor [176,177]. BRCA1-treated BC cells were silenced by genistein therapy, which also caused GPR30 expression to be downregulated, Akt phosphorylation to be inhibited, B1 expression to be downregulated, and cell cycle arrest [178]. Moreover, the therapy increased Nrf2 expression, which decreased ROS levels [178].

The detrimental impact of genistein on DNA methyltransferase may result from the competitive interaction of genistein with hemi-methylated DNA at the catalytic sites of DNA (cytosine-5)-methyltransferase 1, according to in silico studies [179,180]. It has also been demonstrated that genistein activates the Wnt signaling pathway [181]. Treatment with genistein increased the phosphorylation of β-catenin in BC cells, causing it to be limited to the cytoplasm [181]. Wnt signaling and associated genes, like cyclinD1 and cMyc, were also downregulated [181].

The potential mechanism underlying genistein’s anti-BC effect is the downregulation of the estrogen receptor and the vascular endothelial growth factor (VEGFR) that is linked to it [172]. Genistein suppresses the expression of the estrogen receptor and the pathways that lead to it, thus suppressing the expression of VEGFR-2 [182]. In a clinical experiment, genistein or a placebo was administered for one month to 140 women with early-stage BC [183]. This led to an overexpression of genes controlling the cell cycle, including the EGFR2 receptor and tyrosine kinase [183].

MicroRNAs are a class of small non-coding RNAs that are involved in many physiological and pathological processes as post-transcriptional negative regulators. miRNAs function as tumor suppressors or oncogenes in malignant environments, and the dysregulation of miRNA expression has been seen in a wide variety of human malignancies [184].

Studies have been conducted to examine the impact of genistein administration on microRNA expression [185,186]. A study by de la Parra et al., investigated the role of the novel oncogenic microRNA miR-155 in the anticancer effects of genistein in metastatic BC [185]. It was found that genistein inhibited cell viability and induced apoptosis in metastatic MDA-MB-435 and Hs578t BC cells at low concentrations, while miR-155 was downregulated. However, miR-155 levels remained unchanged in MCF-7 cells [185]. In MDA-MB-435 and Hs578t cells, the ectopic production of miR-155 reduces and eliminates the effects of genistein on cell survival, apoptosis, and proapoptotic gene expression [185]. This suggests that the genistein-mediated downregulation of miR-155 contributes to its anticancer effects [185]. The cytoskeleton, a three-dimensional structure in a cell, plays a crucial role in cell shape, movement, and metastatic progression during carcinogenesis [187]. It is regulated by the Rho family of GTPases, RHO, RAC, and cell cycle division 42 (Cdc42) proteins [187]. Recent research reveals that miRNA miR-23b is a central effector of cytoskeletal remodeling, increasing cell–cell interactions, and modulating focal adhesion [187].

### 5.6. Anti-Estrogenic and Estrogenic Properties

Phytoestrogens such isoflavones may function as estrogen agonists in low estrogen environments or as estrogen antagonists in high estrogen environments [188]. Genistein disrupts estrogen binding within molecules, affecting ER-dependent pathways in a dose-dependent manner. It also reverses fadrozole-induced growth inhibition and suppresses Erα mRNA and protein expression in human breast cancer cells [189]. By blocking estrogen, genistein hinders tumor cell proliferation, as it has a stronger affinity for ERβ than ERα, effectively suppressing breast cancer development [65]. Genistein increased c-fos manifestation through ERα and the G protein-coupled receptor equivalent in an ER-independent manner, as demonstrated in ERα-positive MCF7 and ERα-negative SKBR3 breast cancer cells [190]. The expression of the c-fos proto-oncogene could be seen as an initial indicator of estrogenic action in cells [190]. Additionally, when examining the impact of genistein on the inflammation of cancerous cells featuring different ERα and ERβ ratios, it was found that genistein could regulate inflammatory-related genes with the assistance of ER [190].

Genistein has been found to have estrogenic properties in addition to anti-estrogenic and anticancer properties [191]. It functions by using the same basic genetic mechanism on both α and β estrogen receptors; however, it prefers ERβ over ERα [192]. Nonetheless, numerous meta-analyses have not conclusively determined the relationship between genistein and BC [193,194,195]. While some studies have found no link between genistein, menopausal state, and BC, others have suggested that soy consumption may be protective for premenopausal women relative to postmenopausal women [196]. Genistein may be linked to higher survival rates in people who are postmenopausal, ER+, or ER− [65]. It has been seen in certain investigations that BC cells, whether or not estrogen is present, undergo genistein-induced cell death [190]. Daily soy consumption decreased the odds of BC recurrence [188]. In a dose-dependent manner, genistein can disrupt the binding process within estrogen molecules, impacting ER-dependent pathways [197]. It was discovered that inhibiting the expression of ERα mRNA and protein in human BC cells was beneficial [198]. Genistein is capable of preventing the progression of BC because it has a greater affinity for ERβ than ERα [199]. Research has demonstrated that genistein, with the assistance of the ER, can alter gene expression relevant to inflammation [199]. The effects of genistein on the expression of genes and proteins in T47D cells show that focal adhesin kinase, actin, and integrins interact in signaling pathways [199].

### 5.7. Dietary Exposure

A study found that injecting genistein in rats (500 μg genistein/g body weight) during the prepubertal period reduced chemically induced mammary tumorigenesis by 50% [200]. This led to fewer terminal end buds and more lobules of type II [200]. The study also found no significant alterations in fertility, number of offspring, body weight, anogenital distance, vaginal opening, testes descent, estrus cycle, or follicular development [200].

Another study examined the postnatal development of the mammary gland in 80 noninbred Sprague-Dawley virgin rats and the changes induced by 7,12-dimethylbenz[a]anthracene (DMBA) in 60 rats (intragastrically with 10 mg DMBA/100 g body weight) [201]. The mammary gland tree grew, reaching a peak at 21 days old, then decreasing until 63 days old, and slowly decreasing until 84 days old [201]. After DMBA administration, the number of terminal endbuds (TEBs) remained higher than the control animals, and the TEBs became larger and had higher mitotic activities [201]. These TEBs evolved into adenocarcinomas, suggesting that DMBA administration alters the differentiation of TEBs, which leads to AB, which in turn leads to lobules [201]. When genistein was administered to young rats, lobule counts increased while terminal end buds decreased [201,202].

For genistein to exhibit protective effects, researchers found that pre-pubertal and adult exposure to chemically produced BC in genistein-protected rats had to happen between birth and the pre-pubertal stage of mammary gland development [203]. Research indicates that genistein functions as a chemo-preventive drug throughout the pre-pubertal period, which is believed to be akin to adolescence [203]. Based on these investigations, it has been determined that genistein acts on the cell by increasing breast cell differentiation [203,204].

Though genistein has been linked to BC in many studies, further research is needed to fully understand the processes and targets of this link [205]. A study has found that consuming a traditional soy diet in Asian women and men can reduce the incidence of breast and prostate cancers [200]. However, individuals that adopt a Western diet in the US lose this protective effect [200]. The researchers suggest that BC protection in Asian women consuming traditional soy-containing diets may be derived from early exposure to genistein-containing soy [200]. Genistein’s effectiveness in treating colorectal cancer with FOLFOX has demonstrated safety and tolerability despite its usage in research studies, suggesting the possibility for more clinical trials [205]. Comparably, phase II trials examining genistein’s effectiveness in bladder cancer have observed a bimodal effect of the supplement, meaning that it is useful at lower doses and that additional genistein trials in synergy with other medications are warranted [206]. Thus far, genistein has been the subject of three successful clinical trials aimed at treating BC (NCT00244933, NCT00290758, and NCT00099008) [207]. Even at 900 mg per day, soy isoflavone consumption was found to be safe in a phase I double-blind experiment that examined the effects of the supplement on healthy postmenopausal women for 84 days [208]. Despite its strong biological anticancer effects, genistein still has a number of drawbacks, including low solubility (0.01 mg/mL) and low bioavailability [209]. It will be necessary to try new strategies in order to get over these obstacles to its therapeutic use. To obtain further understanding of the effects of genistein, more clinical trials are necessary to investigate its effects on patients with BC. Acute, subchronic, and chronic genistein safety investigations were conducted on Wistar rats [210]. In repeated dose safety trials, the results indicated good tolerance and minimal toxicity [210]. Reduced food intake and body weight gain were noted at 500 mg/kg/day [210]. The LD_50_ of genistein in rats and mice is 500 mg/kg [210]. Hematological results showed elevated reticulocytes and decreased red blood cell characteristics [210]. There were slight alterations in clinical chemistry, but no noteworthy toxicological effects [210]. The study found that rats treated with genistein showed significant changes in organ weight, organ dilation, and ovarian cysts. Male rats showed increased kidney, spleen, adrenal, and testes weights, while female rats showed increased liver, kidney, spleen, ovary, and uterus weights. Histological changes were observed in female reproductive organs, males, and bone, kidneys, heart, liver, and spleen [210]. After 4 weeks of treatment, male rats showed a significant increase in triglycerides, phospholipids, calcium, phosphorus, and chloride, while female rats showed a decrease in uric acid and an increase in total protein. In male rats treated at 500 mg/kg/day, slightly decreased bilirubin, creatinine, cholesterol, glucose, and protein levels, and slightly increased gamma glutamyl transferase were exhibited [210]. In female rats, the total genistein concentration was higher in the liver and kidney, with a higher proportion of free genistein compared to plasma [210]. In the 13 week study, females were lower than males at lower doses and comparable at high doses [210]. In the 52 week study, male rats had higher plasma levels of total genistein than female rats at the mid dose and higher levels at the high dose [210].

**Figure 3 ijms-25-05556-f003:**
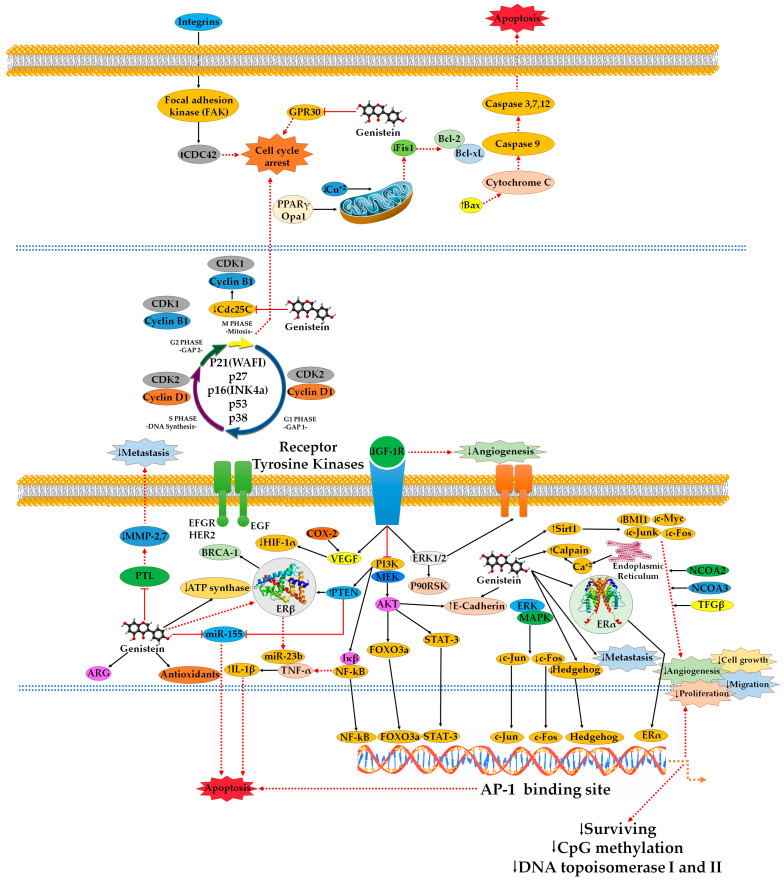
The mechanisms of action of genistein in BC cells. By modifying Bcl-2 family proteins, genistein triggers apoptosis via a mitochondrial-mediated, classical caspase-dependent mechanism. Altering the cycle regulating proteins causes cell cycle arrest. It deactivates the MAPK (ERK1/2) and PI3K/AKT signaling pathways. In addition, genistein controls epigenetic regulation, inhibits angiogenesis, invasion, and cell migration, and modifies the expression of numerous miRNAs. Genistein raises the Bax/Bcl-2 ratio, causing apoptosis through autophagy-dependent pathways, and preventing oxidative stress by altering the expression of antioxidant enzymes. Additionally, genistein inhibits cell proliferation by downregulating CCNG1 GADD45A, NF-κB, Bcl-2, TNFR, ESR1, NCOA2, and NCOA3, and upregulating genes like p53 and CDKN1A. Research conducted in vitro has demonstrated that GNT can reduce tumorigenic processes by upregulating the expression and activity of the GSTP1 and RARβ2 genes. Through the downregulation of the proteins COX, TPA, and EROD, genistein can also inhibit angiogenesis. Fis1 and Opa1 mRNA expression can be decreased by genistein through mitochondrial-dependent pathways, according to in vitro research [211]. In figure, the symbol (↑) indicates an increase and the symbol (↓) indicates a decrease.

## 6. Bioavailability and Metabolism of Genistein

Genistein is an inhibitor of the growth of breast cancer cell lines, MDA-468 (estrogen receptor negative), and MCF-7 and MCF-7-D-40 (estrogen receptor positive) (IC_50_ values ranging from 6.5 to 12.0 micrograms/mL) [212]. The range of 6.5 to 12.0 micrograms/mL is a quantitative measure that indicates how much of genistein is needed to obtain the antitumor effect by 50% in the above cancer cell lines. With its modest molecular weight and advantageous lipophilic characteristic, genistein is an absorbable substance that is almost entirely absorbed in many types of cells [213]. Clinical research on humans indicated that moderate genistein absorption occurred following the oral administration of soy supplements containing high genistein contents [213]. Distinct genistein metabolic pathways were observed in human breast cancer and mammary epithelial cells studied in vitro [213]. In vitro research revealed that genistein was transformed into genistein-7-sulfate and either its hydroxylated or methylated form, with genistein-7-sulfate being the primary metabolite in a number of breast cancer cell lines [214]. It is unclear why there is a difference between human breast tissue obtained in vivo and in vitro cell lines; it could be because breast cancer cell lines express sulfotransferase at a high level [214]. Following high dosages of soy isoflavones, genistein content in breast tissue was comparatively low when compared to plasma, suggesting a modest estrogenic response [214]. The high expression level of sulfotransferase in breast cancer cell lines could be one explanation for the unknown disparity in the genistein metabolism route between in vitro cell lines and in vivo human breast tissue [214]. Following a high dosage of soy isoflavones ingestion (45 mg isoflavones per day for two weeks), genistein concentrations in breast tissue were comparatively low when compared to plasma, indicating a modest estrogenic response in the breast [215]. According to Bolca et al., total genistein concentration in plasma varied from 135 to 2831 nmol/L, whereas total genistein ranged from 92 to 494 pmol/g [216]. Additional analytical measurements revealed that genistein aglycone was responsible for only about 2% of the total genistein in breast tissue, with genistein-7-sulfate being the main metabolite found [216]. According to research by Coldham et al., genistein concentrations in rats’ guts are the greatest (18.5 µg/g), followed by those in their liver (0.98 µg/g), plasma (0.79 µg/g), and other reproductive tissues (uterus, ovary, vagina, and prostate; ranging from 0.12–0.28 µg/g) [217]. While the amounts in other tissues were also above their EC_50_ and compete with estradiol for the activation of the estrogen receptor, the high concentration in the gastrointestinal system tract was sufficient to exert direct antiproliferative effects [217,218]. In rats given an oral dose of 12.5 mg/kg genistein, the majority of the protein accumulated in the stomach (1.83 µg/g), followed by the intestine (1.50 µg/g), liver (1.13 µg/g), kidney (0.41 µg/g), lung (0.27 µg/g), heart (0.23 µg/g), brain (0.1 µg/g), reproductive organs (0.09–0.22 µg/g), and muscle (0.07 µg/g) [219]. In summary, according to ADME tests, genistein has a good intestinal absorption rate; but, without the right formulations, its poor solubility may limit its ability to be absorbed at higher doses [217,218]. The most important factor influencing the low oral bioavailability of genistein is extensive metabolism, which is mostly due to the high expression of efflux transporters, especially Breast Cancer Resistant Protein (BCRP) [217,218]. In order to facilitate enteric and enterohepatic recycling and lower exposure levels, the interaction of metabolic enzymes and efflux transporters is essential for the distribution and removal of genistein. Genistein conjugates’ biodistribution is controlled by efflux transporters, such as BCRP, which influences the pharmacokinetics and bioavailability of the compound [217,218]. Two different recycling processes cause genistein and its conjugates to build up in the liver and gastrointestinal system [217,218]. Clinical trials may be guided by changes in genistein’s in vivo pharmacokinetics and bioavailability caused by gender and sex hormones.

## 7. Conclusions

In conclusion, this article examines the numerous indications that genistein may prevent, delay, or obstruct the development of BC. Preclinical and clinical data demonstrate that genistein has distinct, dose-dependent anti-BC effects that are mediated through multiple distinct molecular pathways. These observations lead to the possibility that genistein is a potent anti-BC drug. However, comprehensive preclinical research and clinical investigations must be planned and carried out. Based on the data currently available, genistein appears to be a promising chemopreventive medication, particularly when it comes to managing drug resistance in cancer patients, which is a clinically significant issue. However, a great deal of effort remains before the therapeutic benefits of genistein can be applied.

## Figures and Tables

**Figure 1 ijms-25-05556-f001:**
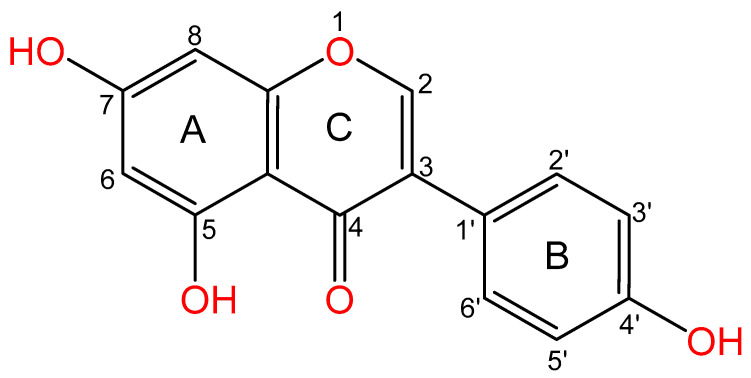
Chemical structure of genistein.

**Figure 2 ijms-25-05556-f002:**
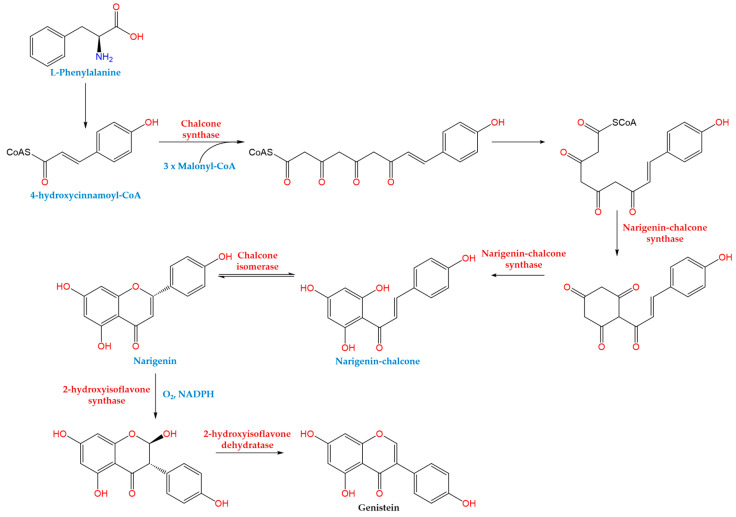
Biosynthetic pathway of genistein.

**Table 1 ijms-25-05556-t001:** Natural occurrences of genistein in 36 plant species. The plant species have been categorized alphabetically.

Plant Species	Plant Species	Plant Species
*Achlys triphylla* [101]	*Calicotome villosa* [102]	*Chaenomeles sinensis* [103]
*Chamaecytisus supinus* [104]	*Cicer arietinum* [105]	*Dalbergia sissoo* [106]
*Eriosema tuberosum* [107]	*Ficus septica* [108]	*Flemingia macrophylla* [108]
*Flemingia paniculata* [109]	*Genista ephedroides* [110]	*Genista lydia* [111]
*Genista sessilifolia* [112]	*Genista tridentata* [113]	*Glycine max* [114,115]
*Grona styracifolia* [116]	*Lupinus albus* [117]	*Lupinus luteus* [118]
*Lupinus polyphyllus* [119]	*Lupinus pubescens* [120]	*Maackia amurensis* [121]
*Mucuna birdwoodiana* [122]	*Neorautanenia amboensis* [123]	*Neorautanenia mitis* [123]
*Prunus avium* [124]	*Prunus cerasus* [58]	*Pueraria candollei* var. *mirifica* [125]
*Pueraria montana* var. *lobata* [126]	*Retama sphaerocarpa* [127]	*Selaginella sinensis* [128]
*Sorbus cuspidata* [129]	*Styphnolobium japonicum* [130]	*Thermopsis lanceolata* [131]
*Trifolium pratense* [132]	*Trifolium resupinatum* [132]	*Trifolium subterraneum* [133]

## Data Availability

All data and analysis are available within the manuscript, or upon request to the corresponding authors.

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
