# Peer review of "Molecular Pathways of Genistein Activity in Breast Cancer Cells"

_ijms, 2024, doi:10.3390/ijms25105556_

Round 1

Reviewer 1 Report

Comments and Suggestions for Authors

The manuscript "Molecular Pathways of Genistein Activity in Breast Cancer Cells" is interesting and would be necessary for natural product researchers.

1) It would be important to have more recent references that confirm the importance of genistein because this would tell us that its biological potential is still a hot topic.

2) The authors should format the figures in the same way.

3) More importantly, authors should discuss topics concerning toxicity, solubility, bio-disponibility or, in general, the ADME properties of genistein, which should be addressed separately.

Author Response

Thank you very much for taking the time to review this manuscript. We gratefully appreciate your valuable comment. Please find the detailed responses below and the corrections highlighted in the resubmitted files.

Reviewer 2 Report

Comments and Suggestions for Authors

In the current review the authors presented the effect of genistein  on breast cancer cells and the integrated chemical biology methods to harness mechanisms of action for important therapeutic advances.

Major comments:

1. You wrote that “genistein is classified as a phytoestrogen and an inhibitor of angiogenesis. Breast cancer treatment consist in administration of estrogen receptor inhibitors (Tamoxifen) or aromatase inhibitors (stop the estrogen production). It was well known that the administration of phytoestrogen is not beneficial in breast cancer.

You wrote too little about the anti-estrogenic properties of genistein. In addition, the information is not convincing.

2. Add please a subchapter concerning the bioavailability and the metabolism of genistein.

Since genistein was synthesized in the laboratory, how much genistein would be needed to obtain the antitumor effect and which route of administration would be used. At pg 11 you wrote only that genisteine presents low bioavailability.

3. pg 11, You wrote that “acute toxicity, chronic toxicity, LD50, should be carried out in the near future”. Since the studies have reached phase 2 level (for bladder cancer in humans) toxicity studies were done for genistein. Please add them. 4. The current article is similar to the article published by  Bhat, S.S.; Prasad, S.K.; Shivamallu, C.; Prasad, K.S.; Syed, A.; Reddy, P.; Cull, C.A.; Amachawadi, R.G. Genistein: A Potent Anti-Breast Cancer Agent. Current Issues in Molecular Biology 2021, 43, 1502-1517, doi:10.3390/cimb43030106. In my opinion, in order to be published in IJMS, it must be reorganized.

Minor comments:

1. In the introduction you wrote too much about polyphenols in general.  

2. Natural occurrences of genistein: it is not of interest for the present topic to write about the antibacterial, antiviral etc  properties of genistein.

3. pg 10,point 5.6. - you wrote: “Daily soy consumption decreased the odds of breast cancer recurrence but did not significantly lower it”. Which is the meaning of this statement?

4. Point 5.7: Dietary exposure and Clinical Studies: -You can’t say “Clinical studies” if you write abut genistein administration to rats -Your subtitle is “dietary exposure…” and you wrote about injecting genistein. It’s not correct. -You must add for all the studies on rats the amount administered and the route of administration.

5. You did not specify anything regarding the solubility of genistein. Please add. At pg 11 you wrote only that genisteine presents low solubility.

6. pg 11, You wrote: “Thus far, genistein has been the subject of three successful clinical trials aimed at treating breast cancer”. Please add them.

7. pg 13, Conclusions:

In my opinion, the expressions “highly treatment option” and “standard cancer treatment option”are not appropriate at this moment. Please reformulate.

Comments on the Quality of English Language

Minor editing of English language is required.

Author Response

Thank you very much for taking the time to review this manuscript. We gratefully appreciate your valuable comment. Please find the detailed responses below and the corrections highlighted in the re-submitted files.

Round 2

Reviewer 1 Report

Comments and Suggestions for Authors

The authors made the necessary corrections and added the required points, so the manuscript is now suitable for publication.

Author Response

Thank you very much for taking the time to review this revised manuscript. 

Reviewer 2 Report

Comments and Suggestions for Authors

1. Add please a subchapter concerning the bioavailability and the metabolism of Genistein.

Since genistein was synthesized in the laboratory, how much genistein would be needed to obtain the antitumor effect and which route of administration would be used.  Este?? At pg 11 you wrote only that genisteine presents low bioavailability.

Bioavailability and Metabolism of Genistein

Genistein is an inhibitor of the growth of breast cancer cell lines (IC50 values ranging from 6.5 to 12.0 micrograms/ml) [214]. With its modest molecular weight and advantageous lipophilic characteristic, genistein is an absorbable substance that is almost entirely absorbed in living things [215].

What kind of breast cancer cell lines? Please add.

What do you mean by living things? Please correct.

You also wrote about “high dosage of soy isoflavones ingestion”. Give please details concerning the high dosage ingestion.

Since genistein was synthesized in the laboratory, how much genistein would be needed to obtain the antitumor effect?

2. Concerning the toxicity studies you wrote “There were slight alterations in clinical chemistry”. Please give details regarding the kidney and liver toxicity.

3.Please improve the quality of Figure 3. 4. The current article is similar to the article published by  Bhat, S.S.; Prasad, S.K.; Shivamallu, C.; Prasad, K.S.; Syed, A.; Reddy, P.; Cull, C.A.; Amachawadi, R.G. Genistein: A Potent Anti-Breast Cancer Agent. Current Issues in Molecular Biology 2021, 43, 1502-1517, doi:10.3390/cimb43030106.

Comments on the Quality of English Language

Minor editing of English language is required.

Author Response

Thank you very much for taking the time to review this revised manuscript. 

Please see the attachment file.
